# Facilitating analysis and dissemination of proteomics data through metadata integration in MaxQuant

Walter Viegener [1,2], Shamil Urazbakhtin [1,2], Daniela Ferretti[1], Jürgen Cox [1] ✉ & Jinqiu Xiao [1] ✉

Metadata plays an essential role in the analysis and dissemination of proteomics data. It annotates sample information for output tables from library searches and displays sample information from data files in public repositories. However, integrating metadata into data analysis can be time-consuming and is not well standardized. Inconsistent metadata formats in public repositories hinder other researchers' ability to reproduce and reuse these public datasets. Here we present the metadata integration in MaxQuant, which provides a user-friendly way to export metadata as SDRF, the standard format that maps sample properties to proteomics data files. We also implemented the annotation of output tables with the SDRF file, enabling users to perform seamless downstream data analysis with annotated output tables. These features provide a simple and standardized approach to creating and leveraging standardized metadata, thereby facilitating data analysis and improving the reusability and reproducibility of public proteomics datasets.

Mass spectrometry-based proteomics is the study of expression, interaction and modification of all proteins in living organisms using mass spectrometry[1]. Advances in instrumentation and data acquisition methods have made it a more powerful tool for studying active biological processes by greatly improving the resolution, sensitivity, reproducibility and throughput of the measurement[2–5]. After the sample measurement, library search is performed to identify and quantify the peptides and proteins[6]. Then output tables that store the library search results need to be annotated with the metadata before the downstream data analysis, due to the fact that the column names of output tables are created based on the data files and may not directly reflect the sample-related properties.

This annotation of output tables could be very cumbersome for studies with large cohorts and complex experimental design. It's also more complicated to annotate studies with multiplexed and fractionated experiments, since there is no direct relation between samples and data files. For multiplexed experiments, multiple samples are included in the same data file[7]. For fractionated samples, multiple data files are related to the same sample[8]. If the metadata are in inconsistent formats across datasets within a single study, such as when reanalyzing multiple public datasets, additional effort would be required for the manual annotation of the output tables[9].

In order to address inconsistent and incomplete metadata in proteomics studies, the Sample and Data Relationship Format (SDRF) was introduced as a standardized format to store the proteomics sample metadata[10]. The SDRF is a tab-delimited text format that describes the relationship between samples and data files in proteomics studies. It consists of sample-related metadata, data file-related metadata and the variables under study. Submitting the SDRF file together with raw files to public repositories like ProteomeXchange[11] is recommended to enhance the reusability and reproducibility of public datasets. Several tools have been developed for exporting[12,13] or reading SDRF[14,15] to promote this standard metadata format.

However, very few proteomics datasets currently include the SDRF file in data submissions[12]. First, creating the SDRF file remains complicated and is not mandatory for data submission. Currently, users need to fill the SDRF file row by row, and each row corresponds

[1]Computational Systems Biochemistry Research Group, Max Planck Institute of Biochemistry, Max Planck Institute of Biochemistry, Martinsried, Germany.
[2]These authors contributed equally: Walter Viegener, Shamil Urazbakhtin. ✉e-mail: cox@biochem.mpg.de; jxiao@biochem.mpg.de

to one sample and data file relationship, therefore for some experimental setups many rows require the entry of repetitive information. For example, multiplexed experiments have repetitive data file properties, and fractionated experiments have repetitive sample properties. Second, users benefit little from creating the SDRF file for their own projects because it doesn't significantly impact the data analysis. Creating the SDRF file requires documenting lots of required data file and sample properties, but most of the data file properties will not be taken into consideration in data analysis. Furthermore, sample properties in the SDRF file currently cannot be used to directly annotate output tables.

In this work, we address the aforementioned challenges by implementing the metadata integration feature in MaxQuant, a widely used software that supports proteomics data analysis from various instruments, quantification techniques and data acquisition modes[16–18]. This feature simplifies the process of creating an SDRF file. We also implement automatic metadata annotation of output tables using Perseus or the provided scripts to further assist users with their own data analysis.

## Results and discussion

### Overview of metadata integration in MaxQuant

In addition to the raw files and FASTA files, metadata integration is implemented in the MaxQuant workflow (Fig. 1). The SDRF file, which is created in the process, consists of all required sample properties, which are mainly filled out by users, as well as all required data file properties, which are extracted automatically by MaxQuant from the input files and user-defined parameters.

The SDRF file could play an essential role in both data dissemination and analysis. First, it could be uploaded to public repositories alongside raw files to improve the reproducibility and reusability of the datasets. Moreover, it can be used to automatically annotate the output tables from MaxQuant for Perseus, as well as other tools using the provided scripts written in Python and R.

### Export metadata as an SDRF file in MaxQuant

Since MaxQuant v2.7.0, the feature to export metadata as an SDRF file has been implemented in the "Metadata" tab (Fig. 2). In MaxQuant, all required data file properties specified by SDRF would be extracted and converted to ontology terms and accession numbers when the SDRF file is written (Table 1). After setting up the parameters for the library search, click the "Refresh" button under the "Metadata" tab, and the corresponding metadata table will appear. To reduce the confusion caused by having too many columns, the metadata table only includes basic data file information and the required sample properties that users need to fill out. An additional column named "Group" is also included, representing the study variable "factor value[group]" in

SDRF. This column doesn't have to follow ontology-based rules, users can define any project-specific treatments or phenotypes that do not belong to one of the required sample properties.

The layout of the metadata table in MaxQuant differs from the SDRF, wherein each row represents one sample and data file relationship. In the metadata table, however, each row represents one sample. Since only sample properties need to be filled in to create an SDRF file in MaxQuant, the layout of metadata table is particularly user-friendly. The exact layout is also automatically generated according to the experimental settings, thereby eliminating the need to fill in repetitive information. For multiplexed dataset, there will be expanded rows corresponding to different labels for each raw file, all labels will be assigned identical data file properties. For fractionated dataset different raw files corresponding to the same sample will be collapsed into one row, these raw files will be assigned identical sample properties. For example, in an experiment with one TMT10 set fractionated into 20 fractions, the SDRF file contains 200 rows. Users only need to fill a 10-row metadata table in MaxQuant to generate the SDRF file. When MaxQuant writes the SDRF file, information from the metadata table will be converted to the SDRF layout and combined with corresponding data file properties.

The metadata table can be filled out in MaxQuant GUI or in the template exported by MaxQuant. There's no maximum limit to the number of rows that can be generated in the metadata table. Users can either edit multiple rows at once in MaxQuant GUI, or fill out the exported template outside of MaxQuant. In the SDRF, some sample properties are required but might not be available for users' projects (e.g. the ancestry category for human samples). Users can leave these columns empty, and MaxQuant will have the option to automatically fill them with "not available". Then the exported SDRF file will have no missing values and can be submitted directly to public repositories without further editing. The SDRF file will be created later as one of the output tables after the library search, or immediately by clicking the "Write SDRF" button (Supplementary Data 1, example of SDRF file generated by MaxQuant).

In summary, the auto-adjusted layout and autofill feature of the metadata table in MaxQuant simplify the process of creating the SDRF file. Automatic extraction of data file properties saves users the effort of collecting and organizing information that rarely impacts their data analysis. It is especially helpful for proteomics facilities to provide a well-annotated SDRF file to clients without the analytical expertise to annotate the data file properties. The possibility to edit multiple rows simultaneously and read the template file improves efficiency especially when handling large datasets. By default, the SDRF file becomes one of the regular tables written in every MaxQuant run. This feature will encourage the deposit of standardized SDRF metadata in public proteomics repositories, helping to improve the reproducibility and reusability of public datasets.

### Annotate the MaxQuant output tables with the SDRF file for downstream data analysis

In MaxQuant, "Experiment" is the unique identifier for each sample, except for multiplexed datasets, in which each sample is represented as a combination of "Experiment" and label. In the SDRF, the source name has the same definition. Therefore, in the MaxQuant-exported SDRF file, the source name is identical to "Experiment" (combined with the label if the dataset is multiplexed). Thus, it can be used to annotate MaxQuant output tables.

Perseus is a widely used software that performs downstream omics data analysis such as pre-processing, statistical analysis and data visualization[19]. Since it was co-developed with MaxQuant, it is especially compatible with MaxQuant output tables. The feature that annotates MaxQuant output tables with the SDRF file has been implemented since Perseus v2.1.5. Clicking "Read SDRF" under "Annot. rows" automatically adds all information from the SDRF file to the

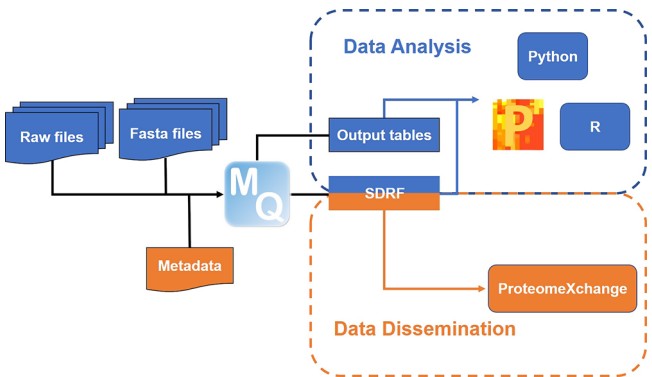

**Fig. 1 | Overview of metadata integration in MaxQuant and its application.** MaxQuant converts metadata to the SDRF file, which can then be used for data analysis and data dissemination. SDRF Sample and Data Relationship Format.

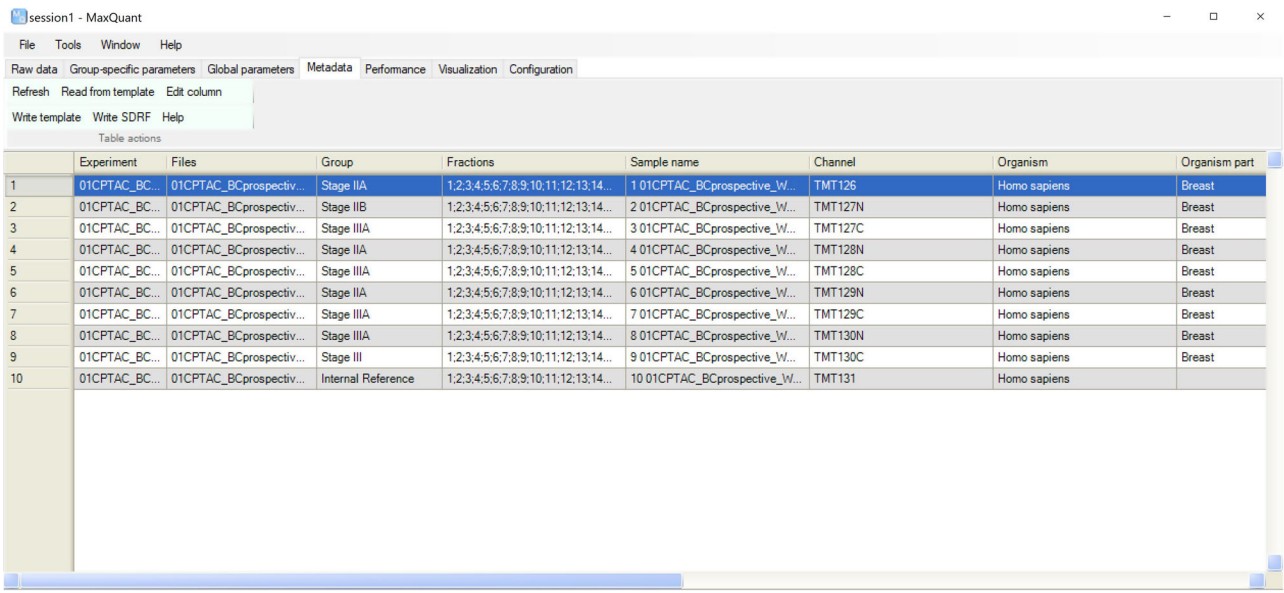

**Fig. 2 | Metadata table in MaxQuant GUI.** The metadata table is automatically generated according to the experimental settings under the 'Metadata' tab. The following screenshot shows the metadata table for a 24-fraction TMT10 dataset.

**Table 1 | Overview of the columns from the SDRF that MaxQuant fills**

| Column name | Source |
|---|---|
| Source name | From "Raw data" -> user-specified "Experiment" |
| Characteristics[organism] | From the fasta file if UniProt database is used, otherwise needs to be filled by users |
| Technology type | Filled with "proteomic profiling by mass spectrometry" |
| Assay name | From file name without file extension |
| Comment[data file] | From file name including file extension |
| Comment[fraction identifier] | From "Raw data" -> user-specified "Fraction" |
| Comment[label] | From "Group-specific parameters" -> "Type" -> user-specified "Type" |
| Comment[cleavage agent details] | From "Group-specific parameters" -> "Digestion" -> user-specified "Enzyme" |
| Comment[instrument] | From instrument information recorded in the raw files |
| Comment[modification parameters] | From "Group-specific parameters" -> "Modifications" -> user-specified "Fixed modifications" and "Variable modifications" |
| Comment[proteomics data acquisition method] | From "Group-specific parameters" -> "Type" -> user-specified "Type" |
| Comment[tool metadata] | MaxQuant with version information |

corresponding intensity columns as annotation rows. By default, "Skip repetitive properties" is selected to only add properties that are not identical between all samples as annotation rows (Fig. 3). With added annotation rows from the SDRF file, users can continue with data filtering, normalization and statistical analysis directly with the annotated output tables. Two scrips (https://github.com/cox-labs/converters), written in Python and R, are provided to convert MaxQuant output tables and the SDRF file into an expression matrix and a sample annotation table. These two files are required by other popular tools for downstream data analysis, such as DESeq2[20], limma[21], and edgeR[22]. As long as the source name is identical to "Experiment" in MaxQuant, SDRF files created by other tools can also be used to annotate output tables with Perseus or can be converted by the provided scripts.

The SDRF was initially introduced to store metadata during data dissemination, which is one of the last steps in finalizing a proteomics study. Here, we extended the application of the SDRF file to output tables annotation, which is a key step in data analysis. As a result, this output table annotation feature will motivate users to create SDRF files because it helps with data analysis. As the SDRF was adapted from the MicroArray Gene Expression Tabular (MAGE-TAB)[23], which is used to

encode metadata annotations for RNA-Seq data[24], SDRF files could also perform as a bridge for multi-omics data integration in the future.

## Methods

### Software development, requirements and usage

Metadata integration is a new feature implemented in MaxQuant v2.7.0 and Perseus v2.1.5. MaxQuant is developed using.NET8 and written in C# programming language. The command-line version can be run on Windows, Linux and macOS (provided that the vendor libraries are available for accessing the raw files). The graphical user interface (GUI) is only available on Windows at the moment. It can be downloaded from https://www.maxquant.org/maxquant. Perseus is developed using.NET8 and written in C# programming language. Its GUI is on Windows only at the moment. It can be downloaded from https://www.maxquant.org/perseus/.

The video tutorials about MaxQuant and Perseus are available at https://www.youtube.com/@MaxQuantChannel. The detailed tutorial on exporting the SDRF file and using it to annotate output tables is provided as PDF in the Supplementary Information. Alternatively, the video tutorial can be viewed on YouTube at https://youtu.be/fHCPOBXXRp8.

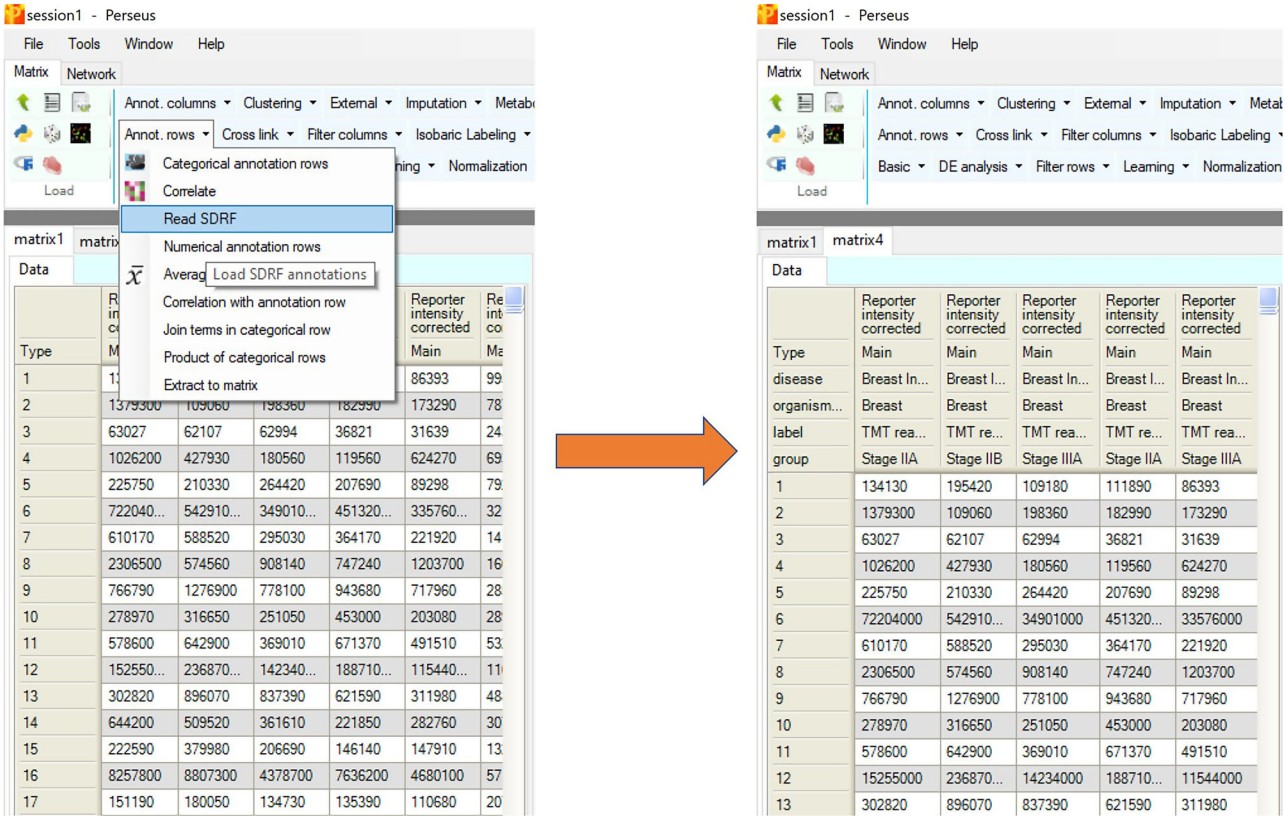

**Fig. 3 | Annotation of the protein groups table with the SDRF file in Perseus.** The protein groups table from a 24-fraction TMT10 dataset is annotated with the SDRF file. "Skip redundant properties" option is selected to only keep the properties that are not identical across all samples.

## Reporting summary

Further information on research design is available in the Nature Portfolio Reporting Summary linked to this article.

## Data availability

No new mass spectrometry proteomics data was generated in the scope of this work. The 24-fraction TMT10 dataset displayed in Figs. 2 and 3 is a subset of the dataset from the CPTAC data portal under accession number S060[25] and was obtained from Proteomic Data Commons (PDC) repository using bash script from https://github.com/esacinc/PDC-Public/tree/master/tools/downloadPDCData.

## Code availability

Both MaxQuant and Perseus are freeware that can be used for all purposes, including commercial use. The partially open-source code was deposited at https://github.com/JurgenCox/mqtools7, under the CC BY-NC-ND 4.0 license. The scripts for annotating the output tables were created using Python v3.12.3 and R v4.5.0, and were deposited at https://github.com/cox-labs/converters, under the CC BY-NC-ND 4.0 license.

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

## Acknowledgements
The authors would like to thank Barbara Steigenberger for her explanation of the metadata management. This work was supported by the German Ministry for Science and Education funding action CLINSPECT-M [FKZ 161L0214E, to J.X.].

## Author contributions
W.V. developed the new features in MaxQuant and Perseus. S.U. performed the tests on the new features. W.V., S.U., D.F., J.C. and J.X. conceptualized the project and designed the new features. J.C. and J.X. supervised the project and drafted the manuscript. All authors have read and approved the final manuscript.

## Funding

## Competing interests
The authors declare no competing interests.
