## [Transparent Peer Review file · Nature Communications]

Facilitating Analysis and Dissemination of Proteomics data through Metadata Integration in MaxQuant

Corresponding Author: Dr Juergen Cox

Version 0:

Reviewer comments:

Reviewer #1

(Remarks to the Author)

In this manuscript, Viegner and colleagues present the integration of metadata in MaxQuant as a progress towards seamless integration of metadata in proteomic data analysis. The implementation is intuitive and will greatly simplify the metadata annotation of MaxQuant data sets while at the same time streamlining downstream interpretation. I therefore support the publication of this work and have only minor comments.

Comments

- The authors are correct that the metadata annotation of proteomic data is lacking and is a barrier towards many applications. More than the missing annotation, research by proteomic repositories has shown that submitters tend to select the first item in the list to submit their data rather than really annotate their data. As the authors point, having the metadata generated as early as possible makes this process much simpler (e.g. 10 vs 200 rows to annotate). The authors can also mention that many proteomic experiments are run by facilities who just ship MaxQuant tables to clients who do not have the analytical expertise to annotate the data. The proposed implementation solves this problem.
- Related to the previous point, I would recommend avoiding prefilling columns with "not available", as this would most likely result in these columns never to be changed.
- Please add a table listing all fields filled by MaxQuant, the source of their content (e.g. filled by user, taken from MaxQuant settings, or from mass spectrometry files). Please also add a short example of SDRF file generated by MaxQuant to the manuscript.

Minor edits

- Please avoid promises on maintenance/development (e.g. it will be continuously maintained and SDRF's involvement in multiomics integration will be implemented in the future), and rather focus on discussing future directions for this field.
- The text could use some proof reading. Please check the tenses and avoid familiar writing.
- For multiplexing dataset, there will be expanded rows correspond to different labels -> multiplexed, corresponding, same next sentence.

(Remarks on code availability)

The code provided is simply a parser for the tables. The implementation itself seems to be closed source. This is disclosed in the text.

Reviewer #2

(Remarks to the Author)

This manuscript presents the implementation of a metadata integration feature in MaxQuant v2.7.0 that supports the creation and export of SDRF (Sample and Data Relationship Format) files, as well as the automatic annotation of output tables using these metadata. This development directly addresses long-standing challenges in metadata standardization within proteomics workflows and offers a practical solution for improving data reusability and reproducibility.

Users can now export SDRF files directly from MaxQuant and include them in submissions to repositories such as ProteomeXchange. Moreover, the integration with Perseus and the provided R/Python scripts enables SDRF-based reanalysis of public datasets. From my perspective, this represents a significant step forward for making proteomics data

more FAIR and further underscores the MaxQuant team's commitment to Open Data and data stewardship best practices.

Before publication, I would encourage the authors to consider addressing the following points to enhance the utility and transparency of the implementation:

- Documentation and Accessibility: As the SDRF specification continues to evolve, how do the authors ensure that MaxQuant users are guided toward the latest SDRF documentation? Would it be possible to integrate contextual help or direct links to the SDRF schema, examples, or tutorials within the MaxQuant interface or documentation?

- Validation and Compliance: Is there a plan to integrate or reference the formal SDRF validator (such as those maintained by the BioSamples/FAIRsharing community) within MaxQuant? Given that SDRF is still expanding—e.g., new ontology terms, experimental designs, and field requirements—how does the MaxQuant team plan to maintain compatibility with the evolving standard? Could support for versioned templates or automatic schema validation be envisioned in future releases?

These suggestions are not intended as critiques of the current implementation, which is already a valuable advancement for the community. Rather, they aim to foster a conversation about the long-term vision for SDRF support in MaxQuant and how it might continue to evolve in alignment with broader community efforts.

(Remarks on code availability)

REVIEWERS' COMMENTS

Reviewer #1 (Remarks to the Author):

In this manuscript, Viegener and colleagues present the integration of metadata in MaxQuant as a progress towards seamless integration of metadata in proteomic data analysis. The implementation is intuitive and will greatly simplify the metadata annotation of MaxQuant data sets while at the same time streamlining downstream interpretation. I therefore support the publication of this work and have only minor comments.

Comments

- The authors are correct that the metadata annotation of proteomic data is lacking and is a barrier towards many applications. More than the missing annotation, research by proteomic repositories has shown that submitters tend to select the first item in the list to submit their data rather than really annotate their data. As the authors point, having the metadata generated as early as possible makes this process much simpler (e.g. 10 vs 200 rows to annotate). The authors can also mention that many proteomic experiments are run by facilities who just ship MaxQuant tables to clients who do not have the analytical expertise to annotate the data. The proposed implementation solves this problem.

That's a good point! Annotating the data file properties for SDRF is difficult for people without analytical expertise, especially given the ontology-based requirements. We addressed this issue in the revised version of the main text.

- Related to the previous point, I would recommend avoiding prefilling columns with "not available", as this would most likely result in these columns never to be changed.

That's true. In future public releases, we'll still keep the possibility to prefill empty cells with "not available," but it will be optional. This will prevent failed submissions due to empty cells in the SDRF file and reduce the user's workload of filling in empty cells.

- Please add a table listing all fields filled by MaxQuant, the source of their content (e.g. filled by user, taken from MaxQuant settings, or from mass spectrometry files). Please also add a short example of SDRF file generated by MaxQuant to the manuscript.

Table and example of SDRF file are added.

Minor edits

- Please avoid promises on maintenance/development (e.g. it will be continuously maintained and SDRF's involvement in multiomics integration will be implemented in the future), and rather focus on discussing future directions for this field.

We've edited the texts accordingly.

- The text could use some proof reading. Please check the tenses and avoid familiar writing.

- For multiplexing dataset, there will be expanded rows correspond to different labels -> multiplexed, corresponding, same next sentence.

We've proof read and made the edits.

Reviewer #1 (Remarks on code availability):

The code provided is simply a parser for the tables. The implementation itself seems to be closed source. This is disclosed in the text.

Reviewer #2 (Remarks to the Author):

This manuscript presents the implementation of a metadata integration feature in MaxQuant v2.7.0 that supports the creation and export of SDRF (Sample and Data Relationship Format) files, as well as the automatic annotation of output tables using these metadata. This development directly addresses long-standing challenges in metadata standardization within proteomics workflows and offers a practical solution for improving data reusability and reproducibility.

Users can now export SDRF files directly from MaxQuant and include them in submissions to repositories such as ProteomeXchange. Moreover, the integration with Perseus and the provided R/Python scripts enables SDRF-based reanalysis of public datasets. From my perspective, this represents a significant step forward for making proteomics data more FAIR and further underscores the MaxQuant team's commitment to Open Data and data stewardship best practices.

Before publication, I would encourage the authors to consider addressing the following points to enhance the utility and transparency of the implementation:

- Documentation and Accessibility: As the SDRF specification continues to evolve, how do the authors ensure that MaxQuant users are guided toward the latest SDRF documentation? Would it be possible to integrate contextual help or direct links to the SDRF schema, examples, or tutorials within the MaxQuant interface or documentation?

In the "Metadata" tab in MaxQuant, we have a "Help" button that links to the GitHub page where we stored the step-by-step manual for exporting the SDRF file and the R/Python script for converting the SDRF file. At the top of the README section (<https://github.com/cox-labs/converters?tab=readme-ov-file#what-is-sdrf>), there is a short description of SDRF and a link to the SDRF documentation which user can easily access.

- Validation and Compliance: Is there a plan to integrate or reference the formal SDRF validator (such as those maintained by the BioSamples/FAIRsharing community) within MaxQuant? Given that SDRF is still expanding—e.g., new ontology terms, experimental designs, and field requirements—how does the MaxQuant team plan to maintain compatibility with the evolving standard? Could support for versioned templates or automatic schema validation be envisioned in future releases?

We will maintain this module to ensure that it keeps pace with future SDRF developments. Instead of providing a reference for users to learn and perform validation themselves, we prefer to perform internal validation to ensure that our exported SDRF file with latest MaxQuant version is already compatible with the evolving standard. We are happy to stay connected with the community and are open to suggestions from them to improve our output and meet the latest standards.

These suggestions are not intended as critiques of the current implementation, which is already a valuable advancement for the community. Rather, they aim to foster a conversation about the long-term vision for SDRF support in MaxQuant and how it might continue to evolve in alignment with broader community efforts.